# Disruptions to youth contraceptive use during COVID-19: Mixed-methods results from Nairobi, Kenya

**Shannon N. Wood**[1,2]*, **Rachel Milkovich**[1], **Mary Thiongo**[3], **Peter Gichangi**[3,4,5], **Meagan E. Byrne**[1,2], **Bianca Devoto**[1], **Philip Anglewicz**[1,2], **Michele R. Decker**[1,2,6,7]

**1** Department of Population, Family and Reproductive Health, Johns Hopkins Bloomberg School of Public Health, Baltimore, Maryland, United States of America, **2** Bill & Melinda Gates Institute for Population and Reproductive Health, Department of Population, Family and Reproductive Health, Johns Hopkins Bloomberg School of Public Health, Baltimore, Maryland, United States of America, **3** International Centre for Reproductive Health-Kenya, Nairobi, Kenya, **4** Department of Public Health and Primary Care, Technical University of Mombasa, Mombasa, Kenya, **5** Faculty of Medicine and Health Sciences, Ghent University, Ghent, Belgium, **6** Center for Public Health and Human Rights, Johns Hopkins Bloomberg School of Public Health, Baltimore, Maryland, United States of America, **7** Johns Hopkins School of Nursing, Baltimore, Maryland, United States of America

* swood@jhu.edu

**Data Availability Statement:** In general, the BSPH IRB is very supportive of making data publicly accessible so long as the risk to individuals and to

## Abstract

Ensuring access to sexual and reproductive health (SRH) services for adolescents is a global priority, given the detrimental health and economic impact of unintended pregnancies. To examine whether and how COVID-19 affected access to SRH services, we use mixed-methods data from young men and women in Nairobi, Kenya to identify those at greatest risk of contraceptive disruptions during COVID-19 restrictions. Analyses utilize cross-sectional data collected from August to October 2020 from an existing cohort of youth aged 16–26. Unadjusted and adjusted logistic regression examined sociodemographic, contraceptive, and COVID-19-related correlates of contraceptive disruption among users of contraception. Qualitative data were collected concurrently via focus group discussions (n = 64, 8 groups) and in-depth interviews (n = 20), with matrices synthesizing emergent challenges to obtaining contraception by gender. Among those using contraception, both young men (40.4%) and young women (34.6%) faced difficulties obtaining contraception during COVID-19. Among young men, difficulty was observed particularly for those unable to meet their basic needs (aOR = 1.60; p = 0.05). Among young women, risk centered around those with multiple partners (aOR = 1.91; p = 0.01), or who procured their method from a hospital (aOR = 1.71; p = 0.04) or clinic (aOR = 2.14; p = 0.03). Qualitative data highlight economic barriers to obtaining contraceptives, namely job loss and limited supply of free methods previously available. Universal access to a variety of contraceptive methods during global health emergencies, including long-acting reversible methods, is an essential priority to help youth avert unintended pregnancies and withstand periods of disruptions to services. Non-judgmental, youth-friendly services must remain accessible throughout the pandemic into the post-COVID-19 period.

groups can be minimized. The IRB approved your plan to allow more limited access to the qualitative data because of the sensitivity of the questions in the cultural setting, respect for the approval by the local IRB, and concern that broader public access could pose risk of stigma to the group that you interviewed. The point of contact is Joan Petit. She can be reached at jhsph.irboffice@jhu.edu. The IRB number for the study in question is 12952. Quantitative data are available upon request from pmadata.org.

**Funding:** This work was supported, in whole or in part, by the Bill & Melinda Gates Foundation [010481]. Under the grant conditions of the Foundation, a Creative Commons Attribution 4.0 Generic License has already been assigned to the Author Accepted Manuscript version that might arise from this submission. The funder was not involved in data collection, manuscript preparation, or the decision to submit this manuscript.

**Competing interests:** The authors have declared that no competing interests exist.

## Introduction

Universal access to sexual and reproductive health (SRH) care for adolescents and young adults is a priority to ensure that youth are able to reach their full potential. Although youth unintended pregnancy rates have decreased throughout sub-Saharan Africa [1, 2], economic and educational disparities [3], coupled with gender-based power imbalances, such as intimate partner violence (IPV) and sexual/contraceptive negotiation [4, 5], remain key inhibitors for youth to obtain their goals. Limited knowledge of contraceptive methods, cost, and social norms stigmatizing contraceptive use outside of marital relationships may further increase difficulty for adolescents and young adults attempting to access contraception [4, 6, 7]. Contraceptive access difficulties are gender-sensitive—young women may be refused contraception due to provider biases, including beliefs that contraceptive use promotes sexualization [8, 9], is linked to infertility [9], and should only be used after women have initiated childbearing [10, 11]. Conversely, masculine gender norms promote young men as knowledgeable about sex, including how to practice safe sex, and as the suppliers of condoms within a service landscape that is generally targeted towards women [12].

High impact practices (HIPs) for youth emphasize non-judgmental service provision, privacy, and availability of a range of free contraceptive methods [13]. However, the availability and/or quality of these services may be altered due to the COVID-19 pandemic [14]. Evidence from recent epidemics, like the Ebola outbreak in West Africa, indicates substantial decreases in contraceptive use due to stock-outs, facility closures, and fear of accessing services [15]. COVID-19 projections have estimated that 15 million additional unintended pregnancies could occur over one year if COVID-related service disruptions affected 10% of women in need of SRH services in low- and middle-income countries [16]. Population-level results from four sub-Saharan African contexts, however, indicate that fear of increased unintended pregnancy may not be grounded in contraceptive use trends—specifically, contraceptive use among women in need increased in the majority of settings, including in urban and rural Kenya [17]. While trends are promising overall, some groups may be at increased risk of contraceptive disruptions, namely, adolescents and nulliparous women [17]. Disruptions to SRH service access during COVID-19 restrictions are of particular concern for urban youth, who rely prominently on coital-dependent contraceptive methods [18], which are more susceptible to service disruptions and less effective in preventing pregnancy.

The first case of COVID-19 in Kenya was detected on March 13, 2020 [19], with school closures, national lockdown, and mandatory curfew immediately following [20]. By late 2020 restrictions were eased, however, in Summer 2021, caseloads began rising again, prompting additional closures [20]. These restrictions, while essential to curbing the spread of COVID-19, could decrease access to essential health services. As Kenya continues to experience waves of COVID-19 cases and restrictions are reinstated, policymakers should draw on lessons learned from the first waves of infection to ensure access to essential services, such as SRH, is maintained while curbing COVID-19 spread. To ensure future policies are responsive to the gendered SRH needs of youth, we used mixed-methods data from young men and women in Nairobi, Kenya to identify those at greatest risk of contraceptive disruptions during mid-COVID-19.

## Methods

### Study design

The present analysis utilizes cross-sectional, mixed-methods data collected mid-pandemic from the Performance Monitoring for Action (PMA) Agile Youth Respondent-Driven

Sampling Survey (YRDSS) cohort. The first round of data collection for this cohort was collected among adolescents and youth ages 15–24 (total n = 1,357; young men n = 690, young women n = 664) in Nairobi, Kenya recruited via respondent-driven sampling (RDS) between June and August 2019. Sampling and study procedures are outlined elsewhere [18, 21].

In 2020, a fully remote follow-up study was conducted with the study cohort to track contraceptive dynamics and assess the gendered impact of COVID-19. Quantitative surveys were conducted by phone in two sessions to limit participant burden: YRDSS Follow-up (young men n = 610, young women n = 613) and Gender/COVID-19 Survey (young men n = 605, young women n = 612). All data for the present analysis were taken from the 2020 follow-up surveys. All survey data was recorded via Open Data Kit (ODK) software. Surveys were conducted in English or Kiswahili per participant preference.

Qualitative data were collected concurrently from August to October 2020. Virtual qualitative methods included focus group discussions (FGDs) with unmarried youth ages 15–24 (n = 64, over eight gender-stratified groups), FGDs with youth-serving stakeholders (n = 32, over four groups), key informant interviews (KIIs) with higher-level family planning stakeholders (n = 12), and in-depth interviews (IDIs) with youth cohort members after completion of follow-up survey (n = 20).

## Analytic sample

The quantitative analytic sample was restricted to those who completed Gender COVID and YRDS follow-up surveys (n = 605 young men, n = 612 young women). Due to skip patterns embedded within the survey, COVID-related contraceptive disruptions were only assessed for those who had ever had sex and were using contraception at follow-up survey (n = 337 young men, n = 347 young women). Contraceptive use was assessed via standard single item: Are you currently doing something or using any method to delay or avoid getting pregnant?" Once restricted to those with complete on COVID-related contraceptive difficulties (<10% missing data), the final analytic sample included 312 young men and 316 young women.

## Measures

The primary dependent variable of interest, contraceptive disruption since COVID-19, was assessed via single multi-response item: "Have you experienced any of the following difficulties in accessing your usual contraceptive methods since the Coronavirus (COVID-19) restrictions began?" Specific contraceptive difficulties comprised: "healthcare facility or doctor's office closed, appointment not possible"; "partner does not approve"; "no transportation to access healthcare services"; "unable to access services because of government restrictions on movement"; "unable to afford healthcare services"; "fear of being infected with COVID-19 at healthcare facilities"; and "other." An affirmative response to any difficulty was classified as a contraceptive disruption since COVID-19.

Independent variables focused on postulated correlates for difficulty using contraception across the socioecological framework, including demographic characteristics (household, partner, individual), contraceptive characteristics, and COVID-19-related factors.

Standard items were used to assess demographic (self-reported household wealth ladder (response 0 (lowest)-10 (highest); analyzed as tertiles), marital status (dichotomous-unmarried vs. married), number of current sexual partners (dichotomous-no partner/one partner vs. multiple partners), age (dichotomous-16-20 vs. 21–26), level of education (categorical-less than secondary, secondary/A level, college/university), parity (dichotomous-nulliparous vs. parous)) and contraceptive characteristics (contraceptive method and place of procurement, both categorical) [22, 23]. Primary contraceptive method was also combined into a categorical

variable to assess method effectiveness: long-acting reversible contraception (intrauterine device (IUD) and implant); short-acting (injectable and pills); barrier or hormonal coital-dependent (emergency contraception (EC), male condoms, and female condoms); and other (rhythm, withdrawal, and herbal pill method).

COVID-19-related items focused on changes to young adults' personal circumstances since the onset of COVID-19 restrictions. Single items assessed change in amount of time spent with partner, control over healthcare, and control over how to spend money; responses of "more", "less", or "no change" categorized the relative frequency of each item in relation to the onset of restrictions. Additionally, a single four-response Likert scale assessed ability to meet basic needs since the onset of COVID-19 restrictions (responses: 1-very able to 4-not able at all); responses categories were collapsed into very able/somewhat able and not very able/not able at all to maximize statistical power.

## Quantitative analysis

Descriptive analyses examined characteristics of the analytic sample and classified contraceptive disruption overall and by gender; significant differences by gender were assessed via design-based F-statistics. Proportions of youth experiencing contraceptive disruption were examined by contraceptive method effectiveness, stratified by gender. Unadjusted and adjusted logistic regression models, using GLM link log and family binomial, examined socio-demographic, contraceptive, and COVID-19-related correlates of contraceptive disruption overall and by gender; adjusted models included correlates with $p<0.1$ from bivariate models (overall or gender stratified), after assessing for multicollinearity.

All analyses were conducted in STATA 16 (College Station, TX) with statistical significance set a priori at $p<0.05$. Sampling weights accommodate the RDS study design using RDS-II (Volz-Heckathorn) weights, post-estimation adjustment based on 2014 Kenya DHS (age, sex, education levels), and modest loss-to-follow-up. All analyses are weighted, with statistical testing accounting for robust standard error clustering by node (recruiter for RDS design at baseline).

## Qualitative data collection and analysis

Youth participants and youth-serving stakeholders were selected to participate in FGDs, IDIs, and KIIs in August to October 2020. Youth participants were identified by community-partnered recruitment through the assistance of local youth organizations; participation in the baseline survey was not a requirement for youth to be selected for the qualitative study. Eligibility criteria were as follows: 1) youth FGDs—15–24 years old, unmarried, living in Nairobi county for at least one year; 2) youth-serving stakeholder FGDs—currently or recently working at a youth-serving organization in a role in direct contact with youth, in Nairobi county, or otherwise knowledgeable about youth health issues in Nairobi; 3) KIIs—health and social service officials and other government officers managing or providing services in the area of adolescent/youth health, social supports, gender equity, and microfinance/economic empowerment; 4) youth IDIs—youth participant in YRDSS cohort with completion of 2020 quantitative survey.

Interviews for youth were conducted via Zoom videoconferencing by trained research assistants. Extensive privacy and confidentiality protections were in place, including password requirements, waiting rooms, no video or chat with other participants, and changing participant display name to an ID. Prior to beginning the interview, the research assistant asked a series of questions to confirm privacy for participants; at this time, a code word was also designated for use, should privacy be compromised during the interview. These virtual sessions

were audio recorded and transcribed from Kiswahili to English. FGDs, IDIs and KIIs followed a semi-structured guide focused on norms and opinions related to youth family planning, relationships, and impacts of COVID-19.

A team of five qualitative researchers coded the 44 transcripts generated by FGDs, KIIs, and IDIs using a deductive approach. An initial codebook was created based on baseline data and existing literature on youth contraceptive behaviors and relationship dynamics. To maximize interrater reliability, all transcripts were dual coded by two researchers. After a set of 2–4 transcripts were coded, the research team met to discuss emergent themes, interpretations of youth experiences, and potential codes to add to the codebook. Retro-coding was used to ensure the quality of analysis and triangulation between transcripts. Matrices were created to synthesize emergent challenges to obtaining contraception during the pandemic (by gender), including COVID-19 concerns/behaviors, government response to the pandemic, and COVID restrictions impacting contraceptive access. For the purpose of this analysis, quotes from youth voices were prioritized to best understand issues youth were directly facing during COVID-19 restrictions.

### Ethical considerations

The Institutional Review Board (IRB) at the Johns Hopkins Bloomberg School of Public Health and the Ethics Review Committee (ERC) at Kenyatta National Hospital/University of Nairobi approved all study procedures. Due to the remote nature of data collection, oral consent activities occurred for both quantitative and qualitative phases, with oral consent responses recorded in ODK (quantitative) or a participant call log (qualitative). The IRB/ERC waived parental consent for this study; therefore, only participant consent was required. Participants' compensation (500 KES/5 USD), decided in consultation with community partners and resident enumerators, was transferred electronically via MPesa. All procedures aligned with ethical best practices for sensitive topics [24, 25], including specialized training, extensive privacy and confidentiality protections (including password protections, auditory privacy screener, no video, and code word in case of interruptions), and provision of resource referrals, specifically for COVID-19 and violence support.

## Results

### Quantitative results

Demographic, contraceptive, and COVID-related factors are presented in Table 1, both overall and by gender. At COVID-19 survey, the majority of the cohort was between ages 21–26 (72.1%) and had completed secondary or A level education (61.3%). Approximately one in ten had married in the past year (11.0%), however, recent marriage was slightly higher for young women (young men = 7.7%; young women = 13.5%; p = 0.09). No significant differences by gender were observed in other demographic characteristics.

Primary contraceptive method and method effectiveness ranged substantially by gender (p<0.001), with 90.8% of young men relying on barrier or hormonal coital-dependent methods, primarily the male condom (86.9%). Barrier or hormonal coital-dependent methods were similarly most widely used for young women (46.1%), however, large proportions also utilized short-acting (30.4%) and long-acting reversible contraceptive methods (21.7%). Most prominent contraceptive methods reported for young women were male condom (38.5%), injectable (25.1%), and implant (18.8%). Place of contraceptive procurement similarly differed by gender (p<0.001), with young men predominantly seeking their method from a pharmacy (56.3%), whereas women were more divided between pharmacies (41.1%) and health centers (29.1%).

**Table 1. Characteristics of young women and young men in Nairobi using contraception at 2020 follow-up survey (n = 628), weighted.**

| | Overall (n = 628) | Young Men (n = 312) | Young Women (n = 316) | p-value across gender |
|---|---|---|---|---|
| | column % | | | |
| *Demographic Characteristics* | | | | |
| Household Wealth | | | | 0.46 |
| Low | 46.7 | 48.9 | 45.1 | |
| Middle | 19.2 | 16.3 | 21.5 | |
| High | 34.0 | 34.8 | 33.4 | |
| Married in the past year | 11.0 | 7.7 | 13.5 | 0.09 |
| Current number of sexual partners | | | | 0.32 |
| No current partner | 7.3 | 8.6 | 6.3 | |
| One current partner | 77.3 | 72.8 | 81.0 | |
| More than one current partner | 15.4 | 18.6 | 12.8 | |
| Age | | | | 0.99 |
| 16–20 | 27.9 | 27.9 | 27.8 | |
| 21–26 | 72.1 | 71.1 | 72.2 | |
| Highest Level of Education Completed | | | | 0.31 |
| Less than secondary | 27.5 | 23.5 | 30.6 | |
| Secondary/ "A" level | 61.3 | 65.8 | 57.8 | |
| College//university | 11.2 | 10.7 | 11.6 | |
| Has children | 11.3 | 11.7 | 11.0 | 0.85 |
| *Contraceptive Characteristics* | | | | |
| Method effectiveness | | | | <0.001 |
| Long-acting reversible contraception | 13.2 | 2.2 | 21.7 | |
| Short-acting | 20.2 | 7.1 | 30.4 | |
| Barrier or hormonal coital-dependent | 65.6 | 90.8 | 46.1 | |
| Other | 1.0 | 0.0 | 1.8 | |
| Main Contraceptive Method | | | | <0.001 |
| Implant | 11.5 | 2.0 | 18.8 | |
| Coil/IUD | 1.7 | 0.0 | 3.0 | |
| Injectable | 15.9 | 4.0 | 25.1 | |
| Oral contraceptive pills | 4.4 | 3.1 | 5.4 | |
| Emergency contraception | 3.9 | 3.2 | 4.4 | |
| Female condom | 2.1 | 0.7 | 3.2 | |
| Herbal Pill Method | 1.0 | 0.0 | 1.8 | |
| Place Obtained Main Contraceptive Method[±] | | | | <0.001 |
| Hospital | 10.7 | 3.7 | 16.1 | |
| Health Center | 23.8 | 17.0 | 29.1 | |
| Clinic/Other Health Provider | 2.6 | 3.2 | 2.0 | |
| Pharmacy | 48.4 | 56.3 | 42.1 | |
| Other | 14.7 | 19.7 | 10.7 | |
| *COVID-related Factors* | | | | |
| Amount of Time Spent with Partner Changed since COVID-19 Restrictions | | | | 0.49 |
| No current partner | 9.3 | 7.5 | 10.8 | |
| Increased | 35.5 | 34.2 | 36.5 | |
| Unchanged | 11.9 | 14.5 | 9.8 | |
| Decreased | 43.3 | 43.9 | 42.9 | |

*(Continued)*

**Table 1.** (Continued)

|  | Overall (n = 628) | Young Men (n = 312) | Young Women (n = 316) | p-value across gender |
|---|---|---|---|---|
|  | column % | | | |
| Change in Control over Healthcare since COVID-19 Restrictions |  |  |  | 0.92 |
| More Control | 40.0 | 39.2 | 40.6 |  |
| Unchanged | 40.4 | 41.7 | 39.4 |  |
| Less Control | 19.6 | 19.1 | 20.0 |  |
| Change in Control over How to Spend Money since COVID-19 Restrictions |  |  |  | 0.04 |
| Does Not Earn Money | 34.1 | 25.9 | 40.5 |  |
| More Control | 33.1 | 40.4 | 27.4 |  |
| Unchanged | 19.9 | 20.4 | 19.5 |  |
| Less Control | 13.0 | 13.3 | 12.7 |  |
| Ability to Meet Basic Needs Since COVID-19 Restrictions |  |  |  | 0.14 |
| Very able/Somewhat able | 48.0 | 52.7 | 44.3 |  |
| Not very able/Not at all able | 52.0 | 47.3 | 55.7 |  |
| Any Contraceptive Procurement Difficulty since COVID-19 Restrictions | 37.1 | 40.4 | 34.6 | 0.29 |
| Specific Contraceptive Procurement Difficulty* |  |  |  |  |
| Healthcare facility or doctor's office closed, appointment not possible | 11.5 | 12.5 | 10.7 | 0.59 |
| Partner does not approve | 0.6 | 0.0 | 1.0 | 0.08 |
| No transportation to access healthcare services | 1.8 | 2.4 | 1.3 | 0.43 |
| Unable to access services because of government restrictions on movement | 10.2 | 15.5 | 6.1 | 0.003 |
| Unable to afford healthcare services | 7.9 | 6.7 | 8.9 | 0.50 |
| Fear of being infected with COVID-19 at healthcare facilities | 19.6 | 20.7 | 18.7 | 0.61 |
| Other | 1.4 | 0.9 | 1.8 | 0.37 |

P-value from design-based F-statistic

Method effectiveness categories: LARC = IUD, implant; short-acting = injectable, pills; hormonal/barrier coital-dependent = EC, male condoms, female condoms; other = rhythm, withdrawal, herbal pill method

*not mutually exclusive

COVID-19 had mixed impact on youth, including time with partner, change in control over healthcare, change in control over how to spend money, and ability to meet basic needs since COVID-19 restrictions; however, only change in control over how to spend money differed significantly by gender (more control: 40.4% of young men vs. 27.4% of young women; p = 0.04).

Over one in three (37.3%) contraceptive users faced contraceptive difficulty during COVID-19 (40.4% young men vs. 34.6% young women). Fear of infection at healthcare facilities (19.6%) and closure of facilities (11.5%) were the most prominent difficulties reported for both genders, however, young men additionally faced disproportionate difficulty accessing services due to government restrictions on movement (15.5% young men vs. 6.1% young women; p = 0.003).

Contraceptive difficulty was most prominent for those using barrier or hormonal coital-dependent methods (42.6% young men; 51.5% young women; Fig 1). For young women, contraceptive difficulty decreased with longer method effectiveness, however, for young men, larger proportions of long-acting users faced contraceptive difficulties than short-acting users.

Table 2 displays bivariate and multivariable logistic regression results, overall and by gender. Within combined gender models, married youth, compared to those unmarried, displayed trend towards decreased contraceptive disruptions (aOR = 0.58; 95% CI = 0.33–1.01;

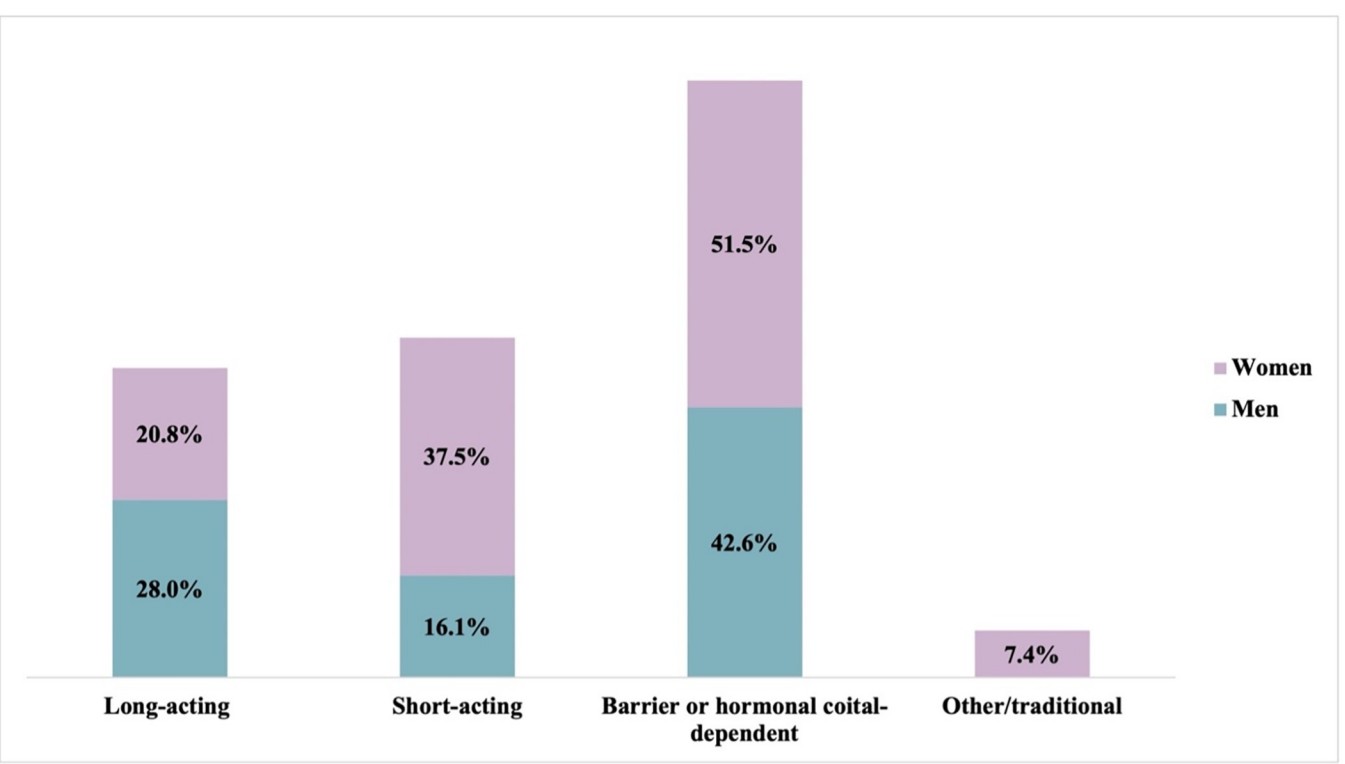

**Fig 1. Difficulty accessing contraception by method effectiveness, by gender.** Method effectiveness: Long-acting reversible contraception = intrauterine device (IUD), implant; short-acting = injectable, pills; hormonal/barrier coital-dependent = emergency contraception.

p = 0.06), whereas multiple partnerships trended towards increased odds of disruption (aOR = 1.34; 95% CI = 0.98–1.83; p = 0.06). When examining COVID-related-factors, youth with more control over healthcare since COVID-19 restrictions, compared to those with unchanged control, showed protection against contraception disruption (aOR = 0.59; 95% CI = 0.42–0.83; p = 0.003).

Young men's contraceptive disruptions centered around COVID-19-related factors. Specifically, young men with higher decision-making control surrounding healthcare reported protection towards contraceptive procurement difficulty (aOR = 0.49; 95% CI = 0.31–0.79; p = 0.004), whereas those with inability to meet basic needs since COVID-19 restrictions faced heighted difficulties (aOR = 1.50; 95% CI = 1.00–2.14; p = 0.05).

Conversely, for young women, procurement difficulties concentrated at the partner-level, where heightened difficulty was seen for those with more than one current partner (aOR = 1.91; 95% CI = 1.20–3.02; p = 0.01), whereas recently marriage trended towards protective effect (aOR = 0.54, 95% CI = 0.27–1.08; p = 0.08). In examining contraceptive factors, those who sought their method from a hospital (aOR = 1.71; 95% CI = 1.02–2.88; p = 0.04) or a clinic/other health provider (aOR = 2.14; 95% CI = 1.09–4.18; p = 0.03) reported increased odds of procurement difficulty, compared to young women who procured their primary method of contraception from a pharmacy.

## Qualitative results

Most young people did not directly experience difficulty in obtaining contraception since the onset of COVID-19; however, those who did faced significant barriers. While economic

**Table 2. Logistic regression examining any difficulty accessing contraception during COVID-19, among contraceptive users at follow-up, by gender, weighted (n = 628).**

| | Overall (n = 628) | | | Young Men (n = 312) | | | Young Women (n = 316) | | |
|---|---|---|---|---|---|---|---|---|---|
| | Difficulty row % | OR (95% CI) | AOR€ (95% CI) | Difficulty row % | OR (95% CI) | AOR† (95% CI) | Difficulty row % | OR (95% CI) | AOR¥ (95% CI) |
| *Demographic Characteristics* | | | | | | | | | |
| Age | | | | | | | | | |
| 16–20 | 41.2 | ref | ref | 47.6 | ref | ref | 36.2 | ref | ref |
| 21–26 | 35.6 | 0.86 (0.64, 1.16) | 0.86 (0.65, 1.13) | 37.6 | 0.79 (0.53, 1.16) | 0.84 (0.58, 1.20) | 34.0 | 0.94 (0.59, 1.48) | 1.02 (0.65, 1.62) |
| Highest Level of Education Completed | | | | | | | | | |
| Less than secondary | 35.9 | ref | ref | 38.3 | ref | ref | 34.5 | ref | ref |
| Secondary/ "A" level | 38.0 | 1.06 (0.73, 1.54) | 0.93 (0.64, 1.36) | 41.6 | 1.09 (0.62, 1.89) | 0.95 (0.54, 1.67) | 34.9 | 1.01 (0.61, 1.69) | 0.99 (0.59, 1.64) |
| College/university | 35.0 | 0.97 (0.59, 1.62) | 0.85 (0.52, 1.39) | 37.6 | 0.98 (0.49, 1.98) | 0.85 (0.42, 1.69) | 33.2 | 0.96 (0.46, 1.99) | 0.93 (0.45, 1.93) |
| Household Wealth | | | | | | | | | |
| Low | 37.4 | ref | | 42.0 | ref | ref | 33.5 | ref | ref |
| Middle | 29.4 | 0.79 (0.52, 1.20) | 0.80 (0.52, 1.23) | 36.6 | 0.87 (0.52, 1.45) | 0.90 (0.55, 1.50) | 25.2 | 0.75 (0.39, 1.43) | 0.74 (0.39, 1.39) |
| High | 41.1 | 1.10 (0.81, 1.49) | 1.04 (0.78, 1.38) | 40.0 | 0.95 (0.63, 1.45) | 0.86 (0.59, 1.28) | 42.0 | 1.25 (0.82, 1.92) | 1.22 (0.80, 1.86) |
| Parity | | | | | | | | | |
| Nulliparous | 37.3 | ref | ref | 40.4 | ref | ref | 34.9 | ref | ref |
| Parous | 35.9 | 0.96 (0.61, 1.52) | 1.07 (0.68, 1.69) | 40.9 | 1.01 (0.52, 1.98) | 0.96 (0.51, 1.81) | 31.7 | 0.91 (0.50, 1.64) | 1.04 (0.58, 1.86) |
| Married in the past year | | | | | | | | | |
| Unmarried | 39.0 | ref | ref | 41.5 | ref | ref | 36.9 | ref | ref |
| Married | 22.2 | **0.57* (0.33, 0.99)** | 0.58± (0.33, 1.01) | 27.7 | 0.67 (0.28, 1.57) | 0.64 (0.27, 1.52) | 19.8 | 0.54± (0.27, 1.08) | 0.54± (0.27, 1.08) |
| Current number of sexual partners | | | | | | | | | |
| No current partner/one partner | 35.2 | ref | ref | 40.9 | ref | ref | 31.0 | ref | ref |
| More than one current partner | 47.8 | 1.36 (0.94, 1.97) | 1.34± (0.98, 1.83) | 38.2 | 0.94 (0.57, 1.54) | 1.06 (0.68, 1.64) | 58.6 | **1.89** (1.16, 3.09)** | **1.91** (1.20, 3.02)** |
| *Contraceptive Characteristics* | | | | | | | | | |
| Main Contraceptive Method | | | | | | | | | |
| Long or short-acting method | 32.7 | ref | ref | 19.9 | ref | ref | 34.6 | ref | ref |
| Coital-dependent | 39.4 | 1.21 (0.89, 1.64) | 1.12 (0.83, 1.51) | 42.6 | 2.26 (0.65, 7.89) | 2.43 (0.70, 8.44) | 34.6 | 1.00 (0.67, 1.49) | 0.89 (0.59, 1.34) |
| Place of Procurement | | | | | | | | | |
| Pharmacy | 37.1 | ref | ref | 45.6 | ref | ref | 28.3 | ref | ref |
| Hospital | 43.7 | 1.18 (0.76, 1.83) | 1.28 (0.88, 1.88) | 42.6 | 0.93 (0.33, 2.67) | 0.90 (0.30, 2.74) | 43.9 | 1.55 (0.90, 2.67) | **1.71* (1.02, 2.88)** |
| Health Center | 34.6 | 0.93 (0.63, 1.37) | 0.99 (0.69, 1.42) | 33.9 | 0.74 (0.41, 1.35) | 0.77 (0.45, 1.33) | 34.9 | 1.23 (0.73, 2.09) | 1.30 (0.77, 2.21) |
| Clinic/Other Health Provider | 42.8 | 1.15 (0.62, 2.16) | 1.16 (0.63, 2.11) | 28.7 | 0.63 (0.19, 2.13) | 0.57 (0.17, 1.90) | 60.7 | **2.14* (1.10, 4.19)** | **2.14* (1.09, 4.18)** |
| Other | 33.2 | 0.89 (0.57, 1.40) | 0.86 (0.56, 1.34) | 31.2 | 0.68 (0.39, 1.19) | 0.70 (0.41, 1.19) | 36.1 | 1.28 (0.63, 2.59) | 1.33 (0.67, 2.65) |
| *COVID-related Factors* | | | | | | | | | |

*(Continued)*

**Table 2.** (Continued)

| | Overall (n = 628) | | | Young Men (n = 312) | | | Young Women (n = 316) | | |
|---|---|---|---|---|---|---|---|---|---|
| **Amount of Time Spent with Partner Changed since COVID-19 Restrictions** | | | | | | | | | |
| Unchanged | 44.5 | ref | ref | 48.1 | ref | ref | 40.3 | ref | ref |
| Increased | 33.0 | 0.74 (0.48, 1.15) | 0.81 (0.52, 1.25) | 32.7 | 0.68 (0.37, 1.25) | 0.74 (0.41, 1.35) | 33.3 | 0.83 (0.46, 1.49) | 0.90 (0.50, 1.61) |
| Decreased | 39.7 | 0.89 (0.58, 1.37) | 0.90 (0.58, 1.41) | 46.1 | 0.96 (0.54, 1.69) | 0.98 (0.57, 1.70) | 34.6 | 0.86 (0.46, 1.58) | 0.82 (0.45, 1.49) |
| No current partner | 31.6 | 0.71 (0.39, 1.29) | 0.77 (0.44, 1.37) | 27.6 | 0.57 (0.17, 1.84) | 0.72 (0.24, 2.12) | 33.8 | 0.84 (0.40, 1.76) | 0.79 (0.38, 1.64) |
| **Change in Control over Healthcare since COVID-19 Restrictions** | | | | | | | | | |
| Unchanged | 42.4 | ref | ref | 50.4 | ref | ref | 35.8 | ref | ref |
| More Control | 24.9 | **0.59**[**] **(0.42, 0.83)** | **0.60**[**] **(0.43, 0.84)** | 24.9 | **0.49**[**] **(0.31, 0.79)** | **0.49**[**] **(0.31, 0.79)** | 25.0 | 0.70 (0.42, 1.15) | 0.70 (0.43, 1.15) |
| Less Control | 51.3 | 1.21 (0.88, 1.66) | 1.11 (0.83, 1.50) | 50.5 | 1.00 (0.64, 1.57) | 1.00 (0.64, 1.57) | 51.8 | 1.44 (0.93, 2.26) | 1.39 (0.89, 2.17) |
| **Change in Control over How to Spend Money since COVID-19 Restrictions** | | | | | | | | | |
| Unchanged | 44.5 | ref | ref | 46.4 | ref | ref | 43.0 | ref | ref |
| More Control | 32.7 | 0.73[±] (0.51, 1.06) | 0.87 (0.61, 1.23) | 35.9 | 0.77 (0.48, 1.25) | 0.93 (0.59, 1.46) | 29.0 | 0.67 (0.38, 1.20) | 0.68 (0.39, 1.20) |
| Less Control | 48.0 | 1.08 (0.73, 1.60) | 1.10 (0.77, 1.58) | 52.6 | 1.13 (0.65, 1.97) | 1.09 (0.66, 1.82) | 44.3 | 1.03 (0.58, 1.82) | 1.10 (0.64, 1.87) |
| Does Not Earn Money | 33.0 | 0.74 (0.51, 1.07) | 0.80 (0.56, 1.15) | 36.5 | 0.79 (0.44, 1.42) | 0.90 (0.52, 1.58) | 31.3 | 0.73 (0.46, 1.16) | 0.71 (0.45, 1.13) |
| **Inability to Meet Basic Needs Since COVID-19 Restrictions** | | | | | | | | | |
| Very able/Somewhat able | 34.6 | ref | ref | 33.3 | ref | ref | 35.8 | ref | ref |
| Not very able/Not at all able | 39.5 | 1.14 (0.86, 1.52) | 1.25 (0.95, 1.64) | 48.4 | 1.46[±] (0.98, 2.16) | **1.50**[*] **(1.00, 2.14)** | 33.6 | 0.94 (0.62, 1.41) | 0.94 (0.63, 1.41) |

[±]p<0.1

[*]p<0.05

[**]p<0.01

[***]p<0.001

[¢]Combined gender model adjusted for marital status, change in control over healthcare, change in control over money

[†]Young men model adjusted for change in control over health (collinearity for ability to meet basic needs)

[¥]Young women model adjusted for marital status only (collinearity for number of partners and contraceptive procurement place)

disparities in Nairobi predated the COVID-19 pandemic, all groups of young people participating in FGDs and IDIs mentioned economic barriers as a predominant and ongoing difficulty in accessing contraception.

Specifically, young people who experienced job loss due to COVID-19 often discussed difficulty accessing contraception. With less available income, an individual facing loss of employment must decide between affording food, rent, and contraception, among other basic needs, which often meant that contraception was deprioritized.

I think when corona came, the outbreak increased poverty in our country. Like all these youths who lost their jobs job due to the virus, they cannot. . .if they were getting like 500 a

day, now if they do not have the money, what they hustle for is for food and not buying those contraceptives, there are those who do not use them.–Young man, 20–24 years

Economic barriers to obtaining contraception were reported by young people patronizing both the public and private sectors. Among young people interviewed, pharmacies tended to be a popular place to access contraception. However, those who generally rely on pharmacies commented on diminished purchasing power at private health facilities due to higher costs relative to the public sector, which were exacerbated by changes in employment and/or income generation.

Coronavirus has affected apart from affecting the economic status of people. . . Let's say, for example, people who don't usually like going to public hospitals to pick these contraceptives they do go to these chemists, chemists in private hospitals to pick them and this is money. So, when it has come because it has ruined the financial status of most people, some have been laid off from their jobs, others even their business had to be closed. Something like that so people don't have money for. . . somebody choses between contraceptives and whatever, food, which will he/she buy? So, it is better to buy food, but contraceptives. . . She/he says eeh!–Young woman, 20–24 years

Covid-19, has affected the youth in terms of getting the contraceptives. . . mostly is, financially because the contraceptives are available even in chemistries (chemists). . . So the main problem is finances because the Covid-19. . . has many people staying at home.–Young man, 15–19 years

Most young people surveyed were dependent on condoms as their primary contraceptive method. Prior to COVID-19, many young people, especially young men, relied on programs supplying free condoms in public places, including schools and hospitals, to obtain contraception. Yet, due to COVID-19 restrictions and generalized fear of COVID-19 exposure, young people who relied on these programs no longer had access to free services.

Now for example with young men, before Covid started in the hospital there was this as in place of keeping this condom. Right now, they don't keep it. So that they can try to avoid that overcrowding or those people who keep coming to hospital.–Female, 20–24 years

I can say that Covid-19 pandemic has affected the access of contraceptives especially to the men. So, for instance you find in, men used to get contraceptives in their washrooms. But currently because the schools are closed. The access to contraception has been cut down.–Young man, 20–24 years

Economic losses and COVID-19 restrictions have also affected NGOs operating programs supplying contraception to young people, specifically those using health centers to distribute contraceptives, therefore causing disruptions to youth dependent on these services.

Now like my area we used to have this as in people of health were coming and giving us services in this health centres, they give us this contraceptives and stuff. But since Corona started those health facilities are not coming.–Young woman, 20–24 years

It has affected because you find you can get most of them were depending on the government to give those NGOs, but right now, there are no gatherings. No one to give them, so you can find it has affected so much, so those who were using are affected, so it can damage them so much.–Young woman, 15–19 years

Beyond economic barriers, young people living with their parents may encounter unique challenges to obtaining contraceptives. Though not specific to COVID-19 restrictions, young women described facing stigma from family and community members for using contraception. Further, as a result of stay-at-home orders, youth worried that their parents would find their contraceptive methods and know that they are sexually active.

> You find that most of the time you spend time with parents they are not going to work because of Covid at the moment, so might go access let's say those contraceptives and carry things like condoms. And come and put them in the house. So, that time that the parent might pass by your room. And comes and find them. So, you find there is fear they will come and find them.–Young man, 15–19 years

Notably, several young women shared in in-depth interviews that they did not experience disruptions to their contraceptive use during COVID-19 because they were using a long-acting method: "I haven't had concern because already I'm in, I'm using it the five, the long term one" (Young woman, 20–24 years).

## Discussion

These mixed-methods results highlight the difficulties that adolescents and young adults overcame to secure contraception during the COVID-19 pandemic. Over one-third (37.3%) of contraceptive users faced hardships in procuring their method of choice, with fear of infection at healthcare facilities (19.6%) and closure of facilities (11.5%) at the fore. Contraceptive procurement difficulties were not gender neutral, with young men disproportionately facing access-related issues due to restrictions on movement (15.5% young men vs. 6.1% young women; p = 0.003). Further, young men's difficulties centered around economic hardship, specifically the inability to meet their basic needs, whereas young women were more likely to report difficulty if they had multiple concurrent partners or sought services from less frequented facilities, such as hospitals or clinics. Gender-responsive, youth-friendly policies must work to ensure continued access to SRH throughout the pandemic. Free provision of services from an array of non-judgmental, youth-friendly providers can combat fears of infection, economic impact, and provider and community stigma.

The economic effects of the COVID-19 pandemic are severe for Nairobi's youth, with approximately half of young men and young women unable to meet their basic needs mid-pandemic. Notably, quantitative and qualitative results diverge surrounding economic impacts on contraceptive access. Qualitative results reveal gender symmetry between economic hardships and difficulty procuring contraception, with interviews detailing pathways between COVID-related job loss and economic trade-offs. Specifically, young men and young women described the difficult risk-benefit decisions made when prioritizing money for food or rent versus contraception or other healthcare needs. Quantitatively, contraceptive procurement difficulties are significant only for young men, who show heightened difficulty with inability to meet basic needs. Divergence of results may point to the resilience of young women and need to prioritize contraception as a basic need given the disproportionate impact of unintended pregnancy that would fall on them. Free SRH services and contraceptive provision can help ensure that youth do not have to trade health and safety for SRH needs.

Unmarried, young women may disproportionately face stigma surrounding contraceptive use, as evidenced by trend towards protection for married (p = 0.08), versus unmarried women, and increased difficulty procuring contraception for young women with more than one current partner (p = 0.01). These results are in line with previous results from across sub-

Saharan Africa [9–11] and Kenya specifically [8] indicating provider biases surrounding young women's contraceptive use. In line with HIPs [13], provider trainings to combat these biases and encourage nonjudgmental provision of services can encourage ultimately increase contraceptive uptake and empower young women to gain control over their reproductive health.

Difficulty accessing contraception during the COVID-19 pandemic differed substantially by method effectiveness, for both young men and young women. Nearly half of barrier or hormonal coital-dependent method users faced difficulties (42.6% young men; 51.5% young women); results were less pronounced for short-acting and long-acting methods, however, 37.5% of young women using short-acting methods (e.g., injectable or pills) indicated access difficulties since COVID-19 restrictions. Heightened difficulties obtaining coital-dependent methods are echoed qualitatively, where young adults describe unavailability of condoms through traditional channels, such as schools and hospitals, throughout the course of the pandemic. While condom use among adolescents was decreasing prior to the COVID-19 pandemic and shorter-term methods, gaining popularity [26], these methods remain an important protection strategies for HIV/STIs among a population with relationship transiency [18]. Long-acting reversible contraceptive methods (e.g., the implant or IUD) are valuable options to help youth avert unintended pregnancies and withstand periods of economic hardship and disruptions to services. These methods, however, must be offered in tandem with a range of alternative options and thorough contraceptive counseling, inclusive of discussions surrounding longevity of protection, potential side effects, and partner involvement, to ensure that youth reproductive autonomy is protected.

Youth narratives reveal difficulty accessing contraception from both public and private healthcare facilities across genders; quantitatively, however, facility-specific disruptions were most prominent for young women accessing contraception from less-frequented facilities. At the earliest stage of the pandemic, higher level facility services may have been shifted from SRH to focus on COVID-19 response [14]. To counteract competing demands at higher level facilities, contraception and SRH services can be reallocated to easier to access points, including pharmacies, over-the-counter, and outreach by community health volunteers (CHVs). Further, these additional points of provision may be essential in combating fears of COVID-19 infection when seeking formal services. Pharmacies specifically are essential for ensuring continued access to coital-dependent methods, including male condoms. Contraception that can be utilized independently of the health system (i.e., self-care methods), such as condoms, DMPA-SC, and multi-pack pills, may be particularly beneficial throughout continued transitions in formal health services.

This mixed-methods analysis was not without limitations. Namely, quantitative analyses were limited to contraceptive users due to skip patterns embedded within the questionnaire and comparison to pre-COVID contraceptive difficulties were not possible. A further understanding of the SRH needs of youth in need of contraception, but who were unable to obtain methods, is necessary to fully assess the range of disruptions Nairobi's youth may have faced during this period of the COVID-19 pandemic. These cross-sectional data were collected August to October 2020, at a time of relative stability in caseloads and may not fully capture heightened disruptions due to recall biases and fluctuation of behaviors over the course of the pandemic. Youth difficulties procuring contraception may have been comparable in pre-COVID periods and should continuously be monitored through subsequent COVID waves and into the post-COVID period. Further, qualitative data were not linked with quantitative data, as inclusion in the cohort was not an eligibility criterion. Not all youth participating in qualitative interviews were using contraception, which may explain divergence of quantitative and qualitative results. While youth did not indicate any difficulties were the virtual data

collection platform and research assistants were trained extensively on monitoring for privacy, it is possible that they were not as forthcoming with their responses given the virtual platform.

The COVID-19 pandemic is far from over in Nairobi and throughout sub-Saharan Africa, as lockdowns and restrictions on movement are reinstated while cases spike. As Kenya enters new waves of infection, policymakers and service providers can glean valuable lessons from the earliest stages of the pandemic. Foremost, media campaigns must balance safety measures while guiding youth to continue to access essentials services. Quality, nonjudgmental contraceptive counseling in the midst of COVID-19 is necessary for youth to select their preferred contraceptive methods and be informed of potential side effects. Free or low-cost services must be made available at places youth frequent most often for contraceptive services (i.e., pharmacy/chemist). As Nairobi reenters COVID-19 restrictions and youth face prolonged periods of economic adversity, continued monitoring of gender-specific contraceptive disruptions is needed to ensure young men and young women are able to access their preferred contraceptive methods; such continuous data collection and monitoring will be imperative to ensure contraceptive disruptions do not alter pregnancy trajectories or hinder youth from achieving their longer-term aspirations.

## Author Contributions

**Conceptualization:** Shannon N. Wood, Mary Thiongo, Peter Gichangi, Philip Anglewicz, Michele R. Decker.

**Data curation:** Mary Thiongo, Peter Gichangi, Bianca Devoto.

**Formal analysis:** Shannon N. Wood, Rachel Milkovich, Meagan E. Byrne.

**Funding acquisition:** Michele R. Decker.

**Investigation:** Mary Thiongo, Peter Gichangi, Michele R. Decker.

**Methodology:** Shannon N. Wood, Peter Gichangi, Bianca Devoto, Philip Anglewicz, Michele R. Decker.

**Project administration:** Mary Thiongo, Peter Gichangi, Meagan E. Byrne, Bianca Devoto.

**Resources:** Philip Anglewicz, Michele R. Decker.

**Supervision:** Mary Thiongo, Peter Gichangi, Meagan E. Byrne, Michele R. Decker.

**Writing – original draft:** Shannon N. Wood, Rachel Milkovich, Michele R. Decker.

**Writing – review & editing:** Shannon N. Wood, Rachel Milkovich, Mary Thiongo, Peter Gichangi, Meagan E. Byrne, Bianca Devoto, Philip Anglewicz, Michele R. Decker.

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
