## [Decision Letter · Decision Letter 0]

31 Aug 2022

PGPH-D-22-01188

Disruptions to youth contraceptive use during COVID-19: mixed-methods results from Nairobi, Kenya

Dear Dr. Wood,

Thank you for submitting your manuscript to PLOS Global Public Health. After careful consideration, we feel that it has merit but does not fully meet PLOS Global Public Health’s publication criteria as it currently stands. Therefore, we invite you to submit a revised version of the manuscript that addresses the points raised during the review process.

Describe if you have done the pr-test. If so, include the information.Describe how was the sampling for the qualitative componentHow was the verbal consent documented (for example, was recorded?)How was decided the compensation for the study participantsHow do you assure confidentiality and privacy using zoom video? Cameras were on?How did you analyse independent variables with list of items like wealth; this is not clearly stated in the text.Waived parental consent was used, so if the respondent with no parent, what would be doneAdd reflection regarding the limitations of inability to compare to pre-COVID contraceptive difficulties or potentially post pandemic.

We look forward to receiving your revised manuscript.

Kind regards,

Camila Gianella Malca, Phd

Academic Editor

Journal Requirements:

1. Please provide a/amend your detailed Financial Disclosure statement. This is published with the article. It must therefore be completed in full sentences and contain the exact wording you wish to be published.

a. Please clarify all sources of funding (financial or material support) for your study. List the grants (with grant number) or organizations (with url) that supported your study, including funding received from your institution. 

b. State the initials, alongside each funding source, of each author to receive each grant.

c. State what role the funders took in the study. If the funders had no role in your study, please state: “The funders had no role in study design, data collection and analysis, decision to publish, or preparation of the manuscript.”

d. If any authors received a salary from any of your funders, please state which authors and which funders.

2. In the online submission form, you indicated that "Quantitative data are available upon request from pmadata.org. To maximize participant privacy, qualitative data are not available.". All PLOS journals now require all data underlying the findings described in their manuscript to be freely available to other researchers, either 1. In a public repository, 2. Within the manuscript itself, or 3. Uploaded as supplementary information.

3. Please provide separate figure files in .tif or .eps format. Please remove the embedded figure from the manuscript file.

4. We have noticed that you have uploaded Supporting Information files, but you have not included a list of legends. Please add a full list of legends for your Supporting Information files after the references list. 

Additional Editor Comments (if provided):

Reviewers' comments:

Reviewer's Responses to Questions

**Comments to the Author**

1. Does this manuscript meet PLOS Global Public Health’s publication criteria? Is the manuscript technically sound, and do the data support the conclusions? The manuscript must describe methodologically and ethically rigorous research with conclusions that are appropriately drawn based on the data presented.

Reviewer #1: Yes

Reviewer #2: Yes

2. Has the statistical analysis been performed appropriately and rigorously?

Reviewer #1: Yes

Reviewer #2: Yes

3. Have the authors made all data underlying the findings in their manuscript fully available (please refer to the Data Availability Statement at the start of the manuscript PDF file)?

Reviewer #1: Yes

Reviewer #2: Yes

4. Is the manuscript presented in an intelligible fashion and written in standard English?

Reviewer #1: Yes

Reviewer #2: Yes

5. Review Comments to the Author

Reviewer #1: Have you done pr-test ,if so include it, How did you include your sample for qualitative part;not clearly stated,how do you assure confidentiality and privacy using zoom video as means of data collection method,how did you analyze independent variables with list of items like wealth;not clearly stated

Waived parental consent was used so if respondent with no parent what would be done?

Make recommendation specific to your finding you were not study out come of contraceptive disruption like the risk of unintended pregnancy,unsafe abortion,further aspiration to use contraception.

Reviewer #2: This manuscript shares findings on the important topic of impact of COVID-19 on access to contraceptives services. The paper is sound with a strong methodology and the results support the conclusions. I support publishing of this paper. I just have a few questions for the authors:

1. How was verbal consent documented.

2. What guided participant compensation of of $5

3. Where there ways to address the limitations pf inability to compare to pre-COVID contraceptive difficulties or potentially

post pandemic.

6. PLOS authors have the option to publish the peer review history of their article (what does this mean?). If published, this will include your full peer review and any attached files.

**Do you want your identity to be public for this peer review?** For information about this choice, including consent withdrawal, please see our Privacy Policy.

Reviewer #1: No

Reviewer #2: No

---

## [Editor Report · Decision Letter 1]

1 Nov 2022

PGPH-D-22-01188R1

Disruptions to youth contraceptive use during COVID-19: mixed-methods results from Nairobi, Kenya

Dear Dr. Wood,

Thank you for submitting your manuscript to PLOS Global Public Health. After careful consideration, we feel that it has merit but does not fully meet PLOS Global Public Health’s publication criteria as it currently stands. Therefore, we invite you to submit a revised version of the manuscript that addresses the points raised during the review process.

We look forward to receiving your revised manuscript.

Kind regards,

Camila Gianella Malca, Phd

Academic Editor

Journal Requirements:

Additional Editor Comments (if provided):

Authors must clarify some procedures, considering ethical principles (confidentiality, not harm, autonomy).

Authors must address the following comments from the reviewers:

Clarify if they have performed a pre-test; if so include it,

Clarify the criteria to include your sample for qualitative part;

Explain how do you assure confidentiality and privacy using zoom video as means of data collection method,

How did you analyze independent variables with list of items like wealth;not clearly stated

Waived parental consent was used so if respondent with no parent what would be done?

Make recommendation specific to your finding you were not study out come of contraceptive disruption like the risk of unintended pregnancy,unsafe abortion,further aspiration to use contraception.
---

## [Editor Report · Decision Letter 2]

19 Jan 2023

Disruptions to youth contraceptive use during COVID-19: mixed-methods results from Nairobi, Kenya

PGPH-D-22-01188R2

Dear Dr. Wood,

We are pleased to inform you that your manuscript 'Disruptions to youth contraceptive use during COVID-19: mixed-methods results from Nairobi, Kenya' has been provisionally accepted for publication in PLOS Global Public Health.

Best regards,

Camila Gianella Malca, Phd

Academic Editor